# Cardiolipin Alterations during Obesity: Exploring Therapeutic Opportunities

**DOI:** 10.3390/biology11111638

**Published:** 2022-11-09

**Authors:** Alexandre Prola, Fanny Pilot-Storck

**Affiliations:** 1Department of Cell Physiology and Metabolism, Faculty of Medicine, University of Geneva, 1211 Geneva, Switzerland; 2Team Relaix, INSERM, IMRB, Université Paris-Est Créteil, F-94010 Créteil, France; 3EnvA, IMRB, F-94700 Maisons-Alfort, France

**Keywords:** mitochondrial inner membrane, OXPHOS, non-alcoholic fatty liver disease, non-alcoholic steatohepatitis, exercise, respiratory coupling

## Abstract

**Simple Summary:**

Understanding the deleterious mechanisms of obesity and ultimately reversing them requires the identification of the key players of metabolic homeostasis altered in this condition. Cardiolipin appears to be one of these. This phospholipid is specifically enriched in mitochondria, the energy powerhouse of cells that converts energy substrates into ATP, the universal energy form of cellular metabolism. Due to its particular structure, cardiolipin confers unique properties to mitochondria and plays a central role in energy production. In this review, we gather a large body of evidence showing the impact of obesity on the quantity and quality of cardiolipin which likely participates in the progressive mitochondrial failure accompanying obesity. However, we also show that different approaches to correct these alterations can improve cellular metabolism and mitigate the consequences of obesity. Furthermore, we report that modulation of cardiolipin could serve as a lever to temporarily increase mitochondrial energy expenditure and induce weight loss. Beyond its deleterious alterations during obesity, cardiolipin could thus offer opportunities in the fight against obesity.

**Abstract:**

Cardiolipin is a specific phospholipid of the mitochondrial inner membrane that participates in many aspects of its organization and function, hence promoting proper mitochondrial ATP production. Here, we review recent data that have investigated alterations of cardiolipin in different tissues in the context of obesity and the related metabolic syndrome. Data relating perturbations of cardiolipin content or composition are accumulating and suggest their involvement in mitochondrial dysfunction in tissues from obese patients. Conversely, cardiolipin modulation is a promising field of investigation in a search for strategies for obesity management. Several ways to restore cardiolipin content, composition or integrity are emerging and may contribute to the improvement of mitochondrial function in tissues facing excessive fat storage. Inversely, reduction of mitochondrial efficiency in a controlled way may increase energy expenditure and help fight against obesity and in this perspective, several options aim at targeting cardiolipin to achieve a mild reduction of mitochondrial coupling. Far from being just a victim of the deleterious consequences of obesity, cardiolipin may ultimately prove to be a possible weapon to fight against obesity in the future.

## 1. Introduction

Human societies are facing an unprecedented epidemic of obesity, accompanied by liver steatosis, accumulated abdominal fat and reduced insulin sensitivity, altogether often leading to a clinical entity called metabolic syndrome [1]. This situation is considered by the World Health Organization (WHO) as a priority for 2030. Indeed, in parallel to the related morbidity burden, at least 2.8 million people worldwide die every year as a result of being overweight.

Metabolic syndrome is a systemic disease accompanied by the dysfunction of multiple organs such as the heart, pancreas, liver and kidney. In search of mechanisms responsible of multi-organ failure, studies have found that nutrient excess induces inflammation and oxidative stress that lead to mitochondrial dysfunction and ultimately organ alteration [2,3,4,5,6,7,8,9,10,11,12]. It has, thus, been proposed that mitochondrial dysfunction is a consequence of obesity, and that restoring mitochondrial function could help preserving organ function.

More recently, another perspective was considered, which places mitochondria at the heart of energy balance and proposes that mitochondrial dysfunction could be a primary driver in the development of metabolic syndrome. Indeed, in response to changes in energy demand and supply, the organism adapts by adjusting both its capacity and efficiency for ATP production, i.e., the ratio of ATP produced by mitochondria per molecule of nutrient. Several publications showed that weight gain in response to overfeeding or weight loss in response to food restriction is highly variable in humans and could be attributed to differences in mitochondrial efficiency in skeletal muscles [13,14,15,16,17]. According to this view, decreasing mitochondrial efficiency could help increasing energy expenditure through the dissipation of excessive ingested energy as heat (for a review, see [18]). 

In recent years, a specific mitochondrial phospholipid named cardiolipin has emerged as an important actor of mitochondrial efficiency, the involvement of which in the metabolic syndrome could offer a lever for new therapeutic opportunities.

The objective of this review is, firstly, to provide an overview of cardiolipin alterations in the whole organism upon obesity. Secondly, we will present recent promising data on cardiolipin that may reveal new opportunities for the management of obesity and metabolic syndrome.

## 2. Mitochondrial Function and Cardiolipin

Mitochondria are major actors in energy production, ion homeostasis, free radicals′ production, and apoptosis, and also work as a signaling platform. Mitochondria produce ATP through the so-called oxidative phosphorylation (OXPHOS) process. According to the chemiosmotic theory, the respiratory chain oxidizes NADH and FADH_2_ to generate an electrochemical gradient and the resulting proton-motive force drives the conversion of ADP into ATP by the ATP-synthase [19]. This process occurs in the mitochondrial inner membrane (mtIM) and requires a specific membrane morphology with invagination of the mtIM into cristae and a precise assembly of respiratory chain subunits into complexes and super-complexes. This highly specific organization necessitates a specific lipid composition of the mtIM and in particular the presence of a unique phospholipid called cardiolipin. Cardiolipin is an anionic molecule, with a dimeric molecular structure comprising three glycerol groups, two phosphate moieties and four esterified fatty acyl chains, which are all bound to a compact polar head group [20,21]. Cardiolipin represents around 10–15% of mitochondrial membranes phospholipids (≈1–5% of the outer membrane, ≈10–15% of the inner membrane) and promotes the formation of mitochondrial cristae (for a review, see [22]). Cardiolipin also plays a crucial role in mitochondrial bioenergetics by favoring the assembly and function of the respiratory chain super complexes [23,24,25,26], the ATP synthase [27,28,29] and the ADP/ATP carrier [30,31,32] and by interacting with kinases such as MtCK [33,34]. Cardiolipin also interacts with numerous other proteins involved in cell death, protein import, calcium transport [35], mitochondrial dynamics and other aspects of mitochondrial biology [19,22,35,36,37,38,39,40,41,42,43,44,45,46,47,48,49]. Cardiolipin could also act as a proton trap [45] and contribute to a local proton circuitry on the mtIM that may optimize proton transfer for efficient mitochondrial coupling [22,46,47,48,49,50]. Cardiolipin that is normally confined at the mtIM is rapidly externalized to the mitochondrial surface in response to the collapse of the mtIM potential. There, cardiolipin constitutes an ‘eat me’ signal for the mitophagic machinery [51,52]. In addition, cardiolipin interacts with cytochrome C and converts this respiratory chain electron transfer unit into a peroxidase that targets the unsaturation sites of cardiolipin acyl chains [53,54]. Cardiolipin species are then hydrolyzed by phospholipase A2, thus generating multiple oxygenated fatty acids, including well-known lipid mediators [55]. Reciprocally, oxygenated cardiolipin triggers cytochrome C release and apoptosis activation.

The main processes and proteins regulated by cardiolipin are illustrated in Figure 1.

### 2.1. Synthesis of Cardiolipin

Cardiolipin synthesis necessitates a complex coordinated sequence of several reactions that are not yet fully understood. Most of our knowledge on cardiolipin synthesis comes from studies on yeast and culture cells; thus, mechanisms governing cardiolipin synthesis flux in vivo would deserve further investigation. Cardiolipin synthesis begins with the production of phosphatidic acid (PA) in mitochondria associated membranes. Glycerol-3-phosphate acyltransferase (GPAT1-4) assembles acyl-CoA and glycerol-3-phosphate molecules to produce lysophosphatidic acid (LPA), then acyltransferase (LPAAT1-4) catalyzes the addition of a second acyl-CoA to produce PA, which is then translocated into the mitochondrial intermembrane space trough unknown mechanisms. Alternatively, PA could be produced directly in the mitochondrial intermembrane space by the acyl-glycerol kinase (AGK). Then, the mtIM resident CDP-DAG synthase TAMM41 converts PA to cytidine diphosphate-DAG (CDP-DAG), which in association with glycerol 3-phosphate is used to form phosphatidylglycerol phosphate (PGP) by the PGP synthase (PGS1) [56,57,58]. PGP is then dephosphorylated into PG by PTPMT1 [59,60], and finally, the cardiolipin synthase (CRLS) assembles PG with a molecule of CDP-DAG to produce cardiolipin [61,62,63]. As opposed to other glycerophospholipids, cardiolipin is, thus, a dimeric molecule comprising four fatty acyl chains, which give it its specific conical shape (for a review, see [22]).

### 2.2. Diversity of Cardiolipin

The de novo synthesis of cardiolipin is followed by a remodeling process, in which cardiolipin undergoes cycles of deacylation and reacylation to acquire a specific fatty acyl composition. Recent methodological progress allowed a precise identification of the acyl chains composing cardiolipin, including 16:0, 16:1, 18:0, 18:1, 18:2, 20:2, 20:3, 20:4, 20:5, 22:2, and 22:6 acyl chains [64,65]. Because of the diversity of the possible acyl chains and the presence of four acyl chains, the diversity of cardiolipin species is virtually very high. However, in mammals, cardiolipin are predominantly composed of 18-carbon unsaturated acyl chains in which linoleic acid (18:2, n-6) or oleic acid (18:1, n-9) represent 85% of the total cardiolipin acyl chains in most tissues, with the noticeable exception of the central nervous system that presents the highest level of cardiolipin diversity with a relatively high proportion of arachidonic (20:4, n-6) and docosahexaenoic acid (22:6, n-3) [64]. In highly oxidative tissues such as the heart and skeletal muscles, tetralinoleoyl-cardiolipin (18:2)4 is the most common cardiolipin species. Because cardiolipin biosynthetic enzymes exhibit no acyl chain specificity [61,62,66], the tissue specificity of cardiolipin acyl chains composition requires a regulated remodeling process that is starting to be understood [65]. First, several isoforms of Ca^2+^-independent phospholipases A2 (iPLA_2_) catalyze the deacylation of nascent cardiolipin to form monolysocardiolipin (for a review, see [67]) that are then reacetylated by acyltransferases. Four different cardiolipin acyltransferases have been identified: ALCAT1 on the ER membrane [68], MLCLAT1 in the mitochondrial matrix [69], Tafazzin in the mitochondrial intermembrane space [70] and, finally, the alpha subunit of the trifunctional protein (TFPa/HADHA) in the mitochondrial matrix, bound to the mtIM [71,72]. In physiological conditions, the vast majority of cardiolipin remodeling is made by Tafazzin. Mutations in the corresponding gene were identified in patients suffering from the Barth syndrome [73], a X-linked recessive mitochondrial disease affecting the heart, skeletal muscle and neutrophil leucocytes [70]. In the following years, Tafazzin was proved to be a major acyltransferase regulating cardiolipin remodeling [74,75]. These data highlight the crucial role of mature cardiolipin composition for mitochondrial function. Accordingly, cardiolipin remodeling is a prerequisite for concentrating OXPHOS subunits and for optimal production of ATP [76].

Recently, the group of Marcus Keller performed multiomics analysis on 15 murine tissues and used artificial neural network, which showed that cardiolipin diversity does not depend on the differential expression of genes involved in cardiolipin synthesis or remodeling, but rather depends on the phospholipid environment [64]. In particular, they identified a strong correlation between the level of a specific fatty acid (and in particular 18:2) in the phospholipid pool, and its level in cardiolipin. They confirmed in vitro that the availability of 18:2 fatty acid in the medium was inversely correlated with the incorporation of other fatty acid in cardiolipin. In particular, these data suggest that the balance in 18:1 and 18:2 acyl chains in phospholipids is of major importance for cardiolipin composition. The authors also suggest that the expression of enzymes regulating cardiolipin biosynthesis or remodeling is unlikely rate-limiting under physiological conditions. In conclusion, fatty acid composition of the medium or in the diet likely plays a crucial role in cardiolipin diversity [77,78]. Accordingly, numerous publications have shown a direct link between fatty acid composition in diet and cardiolipin acyl chain composition (for a review, see [79]). Correlation of cardiolipin acyl groups with fatty acid availability may, thus, represent a mechanism of adaptation to optimize mitochondria efficiency to their environment [80], as exemplified by the cold-driven cardiolipin remodeling that allows thermogenesis in fat [81].

In this context, we will summarize available data regarding the consequences of obesity on cardiolipin composition and present recent publications that suggest that targeting cardiolipin in obesity could represent new therapeutic avenues.

## 3. Cardiolipin Alterations in Obesity

Obesity has been now clearly associated with alteration of mitochondrial function (for a review, see [2]). Given the fact that cardiolipin acyl chain composition is highly dependent on dietary lipids [82,83,84,85,86] and that cardiolipin are mostly composed of unsaturated acyl chains that are very sensitive to oxidative stress [55,87,88,89], it can be hypothesized that alterations in cardiolipin content, acyl chain composition or oxidation may participate to obesity-induced mitochondrial dysfunction. In this section, we will summarize the available data on cardiolipin in obese patients or animal models.

### 3.1. Liver

Obesity is associated with a spectrum of liver abnormalities, known as non-alcoholic fatty liver disease (NAFLD) that is characterized by an increase in intrahepatic triglyceride content with or without inflammation and fibrosis. NAFLD has become an important public health problem because of its high prevalence and potential progression to severe liver disease [90]. Numerous publications evidenced mitochondrial abnormalities in NAFLD (for a review, see [91]).

Liver biopsies from human patients showed a global increase in cardiolipin content both in NAFLD and the more severe non-alcoholic steatohepatitis (NASH) [6]. Acylcarnitine level was unchanged in NAFLD, but accumulated upon the progression toward NASH. The authors suggested that this evolution could reflect a stimulation of hepatic mitochondrial capacity in individuals with NAFLD to protect against its progression. However, this increased respiration may not be sufficient at long term and may lead to excessive oxidative stress [92]. Progression of mitochondrial dysfunction may, thus, lead to the accumulation of acylcarnitine and the development of NASH. NASH/NAFLD was also associated with a remodeling of cardiolipin. Indeed, acyl chain content showed an increase of mono-unsaturated fatty acids (MUFA) and a decrease of di-unsaturated fatty acids (DUFA; including (18:2)4 cardiolipin), with no change in saturated fatty acids (SFA) and poly-unsaturated fatty acids (PUFA). Results obtained in samples from human patients suffering from NAFLD and NASH were corroborated by data on liver from rats fed a high-fat diet (HFD) [93,94]. Depending on the studies, HFD-mediated fatty liver in mice was associated with a decrease of (18:2)4 cardiolipin and an increase of oxidized cardiolipin and monolysocardiolipin [95], or with an increase of (18:2)4 cardiolipin at the detrimental of other cardiolipin species and impaired respiratory complex activity [96]. In both HFD-treated mice or *db*/*db* obese mice, pathological remodeling of cardiolipin in the liver was associated with an increased expression of the acyl-transferase ALCAT1 [97]. To sum up, beyond some contradictory observations, all studies converge to an upregulation of cardiolipin content or cardiolipin remodeling in the liver following lipid accumulation, likely as an attempt to counteract liver failure. While this may preserve mitochondrial function in the early stage of the disease, progression of steatosis finally leads to mitochondrial respiration decompensation, ROS production increase and cardiolipin species remodeling. Altogether, this contributes to mitochondrial dysfunction and transition toward NASH.

### 3.2. Kidney

In the kidney, excess of fatty acids that cannot be oxidized by mitochondria are deposited as lipid droplets, which are associated with glomerulosclerosis, interstitial fibrosis and albuminuria, resulting in high risk of developing chronic kidney disease [98]. Transcriptomic analysis of renal biopsy samples from patients with diabetic nephropathy or mice fed HFD revealed an increased expression of genes involved in lipid uptake and a decreased expression of genes supporting fatty acid oxidation [99,100]. Importantly, there was a significant correlation between the expression pattern of genes involved in lipid metabolism and the glomerular filtration rate in diabetic nephropathy. Similarly to the liver, the kidney may first adapt to preserve normal mitochondrial respiratory function, despite lipid overload. However, the resulting increase in mitochondrial ROS production may ultimately impair mitochondrial respiration in the long term [9]. Peroxidation of cardiolipin has been involved in this context [10], as well as alterations of cardiolipin toward a decrease of total cardiolipin and an increase of lysocardiolipin, which were observed in a swine model of metabolic syndrome [101]. Another study using imaging mass spectrometry also confirmed a decrease of cardiolipin content in the kidney of HFD-treated mice [102].

### 3.3. Heart

Obesity is associated with an elevation of cardiovascular risk, notably because of dyslipidemia, type 2 diabetes, hypertension and sleep disorders [103].

To date, there are no data reporting on the impact of obesity or metabolic syndrome on cardiac cardiolipin content and composition in human patients. In mice fed an HFD, the total level of cardiolipin was unchanged, but a remodeling of cardiolipin acyl chains was reported, although the level of (18:2)4 cardiolipin, which represents more than 85% of cardiolipin, was unchanged and mitochondrial function was preserved [96]. Oppositely, in leptin-deficient mice (*ob/ob*), the total level of myocardial cardiolipin was decreased and associated with a remodeling and a decrease of (18:2)4 cardiolipin [104]. 

### 3.4. Nervous System

The nervous system may act in two ways in obesity. On the one hand, energy homeostasis is regulated by food intake centers in the central nervous system (CNS), which receives information from peripheral organs and sends out efferent neural and hormonal signals to maintain the balance between food intake and energy expenditure [105,106]. Unbalance between these two parameters reflects a CNS dysfunction, leading to excessive energy intake and obesity [107]. On the other hand, obesity has been identified as a risk factor for the development of neurological disorders, such as mild cognitive impairment, altered hippocampal function and Alzheimer′s type dementia [108]. Such diseases have been associated with alteration in calcium regulation and oxidative stress that result in mitochondrial dysfunction in the CNS or peripheric nervous system (PNS; for a review, see [108]).

While defects in cardiolipin content and remodeling have been observed in many neurological disorders (for a review, see [109]), we found only one study that investigated the effects of obesity on cardiolipin in the CNS. In leptin receptor-deficient mice (*db*/*db)*, cardiolipin content was increased in the CNS (spinal cord and brain stem) and unchanged in PNS (sciatic nerve and dorsal root ganglia). In both CNS and PNS, a profound remodeling of cardiolipin was visible, with a noticeable decrease of (18:2)4 cardiolipin in dorsal root ganglia. However, we must keep in mind that oppositely to what is observed in other tissues, 18:2-18:2-18:2-18:1, 18:2-18:2-18:1-18:1 and (18:1)4 cardiolipin, but not (18:2)4 cardiolipin, are predominant in the nervous system. While mitochondrial respiration was unchanged in the spinal cord, it was impaired in the sciatic nerve of these mice [5].

### 3.5. Adipose Tissue

Two types of adipose tissue can be distinguished: the white adipose tissue (WAT) that stores excess energy as triglycerides and the brown adipose tissue (BAT) that dissipates energy through the production of heat in a process called non-shivering thermogenesis. The brown color of BAT comes from a high density of capillaries and mitochondria, which are able to oxidize fat and produce heat thanks to the specific expression of the UCP1 protein (also called thermogenin). UCP1 protein is inserted in the mtIM and dissipates the proton gradient generated by the respiratory chain, which produces heat instead of ATP [110]. Interestingly, cardiolipin binds to and stabilizes UCP1 structure [111]. In WAT, cardiolipin was detected neither in a reference lipidome nor in obese patients [112]. In human BAT, an association was found between the expression of CRLS and whole-body metabolism, but no lipidomic data are currently available [81].

White adipocytes can convert into an intermediate phenotype, called beige adipocytes that are able to produce heat and contribute to whole body energy expenditure (for a review, see [113]). Cold exposure is a key stimulus stimulating white adipocytes “beiging” that results in important modifications of the lipidome [81,114]. In particular, cardiolipin level is increased in both WAT and BAT in response to cold exposure and the subsequent increase in CRLS expression. Interestingly, CRLS expression correlates with UCP1 expression and its overexpression is sufficient to increase fat thermogenesis in both murine and human cells [81]. Accordingly, knockout of CRLS in adipose tissues leads to an almost complete ablation of cardiolipin and blunts the response to cold exposure.

### 3.6. Skeletal Muscle

Skeletal muscles represent the first energy sink of the body because of their capacity to produce ATP energy required to achieve mechanical contraction [115]. The decrease of locomotor activity often reported in obese patients leads to a relative reduction of energy expenditure, which also contributes to weight gain [116]. In addition, inflammation, insulin resistance and lipotoxicity all participate to mitochondrial dysfunction, which, in association with a reduction of muscle mass, are referred to as “sarcopenic obesity” [117,118].

Recently, an analysis of skeletal muscle from obese patients revealed a global decrease in cardiolipin content, without major modifications of fatty acid composition [119]. As in skeletal muscle, cardiolipin content is the best marker for mitochondrial mass [120], this observed decrease of cardiolipin content, thus, likely reflects a reduction in mitochondrial mass [8], which may result from the general reduction of locomotor activity in these patients [116]. In addition, because of its role as a source of lipid mediators [55], the decreased level of cardiolipin may impair oxygenated lipid signaling of the muscle. In isolated mitochondria from rat muscle, no modification of cardiolipin level and only minor modifications of acyl chain composition were observed after 65 days of HFD [121]. As cardiolipin content and mitochondrial mass in muscle are strongly correlated with muscle activity [122], these publications suggest that the reduction of cardiolipin content in obese patients likely correlates with their reduced locomotor activity.

A summary of all these changes in cardiolipin content and composition during obesity is illustrated in Figure 2. 

Taken together, these studies show that depending on the tissue, the content of cardiolipin may increase or decrease in response to obesity, and are certainly connected with changes in mitochondrial mass.

In parallel, a remodeling of cardiolipin species is observed in several tissues, with the noticeable exception of cardiac and skeletal muscles. Since a direct link between fatty acid composition in diet and cardiolipin acyl chain composition has been observed in several publications (for a review, see [79]), this remodeling may reflect changes in the available fatty acids during obesity. On the contrary, native cardiolipin species undergo an intense remodeling in cardiac and skeletal muscle that results in a strong predominance of tetralinoleoyl-cardiolipin (18:2)4. The independence of cardiolipin species composition from obesity likely indicates that the availability in fatty acids is secondary in muscles compared to their endogenous CL remodeling mechanisms.

The role of the molecular diversity of cardiolipin species is still an open question and represents an exciting domain for future investigation. In the context of obesity, the consequences of cardiolipin remodeling remains to be addressed as cardiolipin could represent a new target for the treatment of obesity.

## 4. Targeting Cardiolipin to Treat Obesity

Given the modifications of cardiolipin content and acyl chain composition in response to obesity and their interplay with mitochondrial dysfunction, it has been proposed that targeting cardiolipin could be an interesting option to mitigate obesity consequences or even to treat obesity. Two different strategies are envisioned: (i) approaches aiming at increasing cardiolipin content or restoring cardiolipin composition or integrity towards oxidative stress to alleviate obesity comorbidities; and (ii) approaches targeting cardiolipin to increase mitochondrial oxidation of energetic substrates to increase energy expenditure and induce weight loss.

### 4.1. Mitigation of Obesity-Associated Comorbidities

Mitochondrial dysfunction is part of the mechanisms leading to comorbidities associated with obesity and metabolic syndrome [2]. In addition, data have accumulated regarding the role of cardiolipin in obese patients; thus, it has been proposed that restoring cardiolipin content or integrity could be an interesting strategy to prevent the development of obesity-associated comorbidities. Exercise is a first-line approach in this way, but genetic and pharmacological proofs of concept are also emerging.

#### 4.1.1. Exercise

While the efficacy of exercise to induce long-term body weight loss is still debated [123,124,125], it is now established that exercise, even with no change of body weight, can improve insulin sensitivity and reduce health risks for obese people [126,127,128]. Several publications highlight the positive effects of exercise on cardiolipin.

In obese women, 12 weeks of exercise that consisted in combination of aerobic and resistance training, 4 times per week, improved mitochondrial respiration and increased cardiolipin content, as a reflection of mitochondrial mass [129]. In a study that evaluated the combination of bariatric surgery and exercise on a large cohort of obese patients (n = 51 controls and n = 50 patients submitted to exercise for 10 weeks, 157 min per week, with all patients subjected to bariatric surgery), bariatric surgery did not modify cardiolipin species, while training induced an increase of (18:2)4 cardiolipin proportion in muscle and an improvement of mitochondrial respiratory capacities [87]. In another cohort, obese-diabetic patients were enrolled in a protocol that consisted in 16–20 weeks of a specific diet (25% reduction of calorie intake) combined with exercise of moderate intensity (sessions of 30–40 min at 60–70% of maximal heart rate on most days of the week) [130]. The authors reported that the weight loss and improvement in muscle function and insulin sensitivity were associated with an increased mitochondrial mass and cardiolipin content. The study of another cohort of obese-diabetic patients submitted to a similar exercise protocol confirmed an increase in both mitochondria and cardiolipin content in skeletal muscle and further reported a modest but significant relative increase in (18-2)4 cardiolipin content [131].

Taken together, these studies show a key role of exercise on cardiolipin content as a result of the stimulation of mitochondrial biogenesis. Interestingly, this effect may lead to health improvement independently of weight loss. Whether exercise also protects cardiolipin from peroxidation or altered remodeling remains to be investigated.

In parallel, several publications have provided proof of concepts of a positive effect of cardiolipin content or composition modulation, independently of mitochondrial content increase, through genetic or pharmacological approaches.

#### 4.1.2. Modulation of the Cardiolipin Synthetic Pathway

Several studies have investigated the effects of modulating cardiolipin synthesis pathway in obesity and metabolic syndrome.

For example, Tu et al. studied the role of CRLS1, which is involved in the first step of cardiolipin synthesis, and they discovered that CRLS1 is significantly downregulated in liver of mice fed an HFD or *ob*/*ob* mice and that liver-specific deletion of CRLS1 aggravates HFD-induced insulin resistance and hepatic steatosis [132]. Transcriptomic analyses identified ATF3 as a target of CRLS1 and in vitro approaches revealed that CRLS1 both inhibits ATF3 expression and reduces lipogenesis and inflammation of hepatocytes. The authors proposed that the inhibition of ATF3 expression may result from a CRLS1-mediated increase in cardiolipin synthesis and a subsequent decrease of mitochondrial ROS production. Further investigations are warranted to confirm this hypothesis and the potential interest of manipulating cardiolipin synthesis through CRLS1 expression.

In the same line, Sustarsic et al. found a positive correlation between CRLS1 expression in human adipocytes and insulin sensitivity [81]. They also showed that the overexpression of CRLS1 increases mitochondrial respiration and energy expenditure of human adipocytes.

In parallel, Yeung et al. reported that the protective effect of dulaglutide, a glucagon-like peptide-1 receptor agonist, on kidney function of HFD-induced obese mice, is associated with an increased expression of genes involved in cardiolipin synthesis and remodeling such as *Crls* and *Taz*, resulting in an increase of cardiolipin content [133].

#### 4.1.3. Protection of Cardiolipin against Peroxidation

Using a mouse model, Ye et al. showed that the overexpression of the catalase antioxidant enzyme that catalyzes the conversion of the ROS hydrogen peroxide into oxygen and water molecules is sufficient to preserve cardiomyocytes contractile function [134]. Targeting ROS, thus, appears to be an interesting way to mitigate obesity-associated comorbidities. In the same line, Szeto et al. proposed that protecting cardiolipin from the increase of mitochondrial ROS production of obese mice could protect kidney function. To test this hypothesis, they used SS-31 (also known as elamipretide), a peptide that protects cardiolipin peroxidation through partly unknown mechanisms [135]. Treatment with SS-31 was shown to protect mitochondrial function and prevent HFD-induced glomerulopathy and proximal tubular injury [10]. Investigation of the precise effects of HFD and SS-31 on cardiolipin content, remodeling and peroxidation would be of great interest in this context. Several other mitochondria-targeted inhibitors of cardiolipin oxidation have been developed to prevent apoptosis activation. Promising compounds comprise new electron scavengers able to prevent the production of H_2_O_2_, cationic derivatives of plastoquinone or imidazole-substituted fatty acids that prevent CL oxidation [89,136,137,138,139,140,141,142]. The evaluation of their potential benefits in the context of obesity will be very interesting.

Together, these studies reveal that restoring cardiolipin content, per se or in parallel to mitochondrial mass, as well as cardiolipin integrity towards ROS-induced injury may be an interesting way to preserve mitochondrial function to mitigate obesity-induced comorbidities.

### 4.2. Promotion of Weight Loss through Increased Energy Expenditure

Basically, obesity results from an imbalance between energy intake and energy expenditure. Prevention or treatment of obesity-related disorders include a low-energy diet and beneficial effects of regular exercise, yet the required level of energy restriction is hard to cope with in the long-term, as exercise compliance is generally low in obese people and appetite is often unfortunately stimulated by exercise, canceling its benefit in this aim [123,124,125]. In this context, an appealing strategy to combat obesity and restore insulin sensitivity consists in increasing lipid oxidation through a controlled reduction of cellular energetic efficiency based on mitochondrial uncoupling [18,143]. To reach this goal, efforts were focused on the development of uncoupling compounds, such as dinitrophenol (DNP), or strategies based on uncoupling proteins such as UCP1 or the ATP/ADP carrier [18,144,145,146]. However, increasing substrates consumption at the expense of ATP production may be deleterious, in particular for heart function and heat control [147]. Promising strategies may come from the modulation of mitochondrial coupling restricted to specific organs, such as the liver or muscle [46,148].

In addition to proteins, lipids have been involved in mitochondrial coupling efficiency. In particular, several publications suggest that targeting cardiolipin synthesis or remodeling could provide an opportunity to reduce mitochondrial efficiency and increase energy expenditure for the management of obesity and metabolic syndrome.

Given the observation that the expression of the acyl-transferase ALCAT1 is increased in liver, heart, and skeletal muscle of HFD-treated mice and *db*/*db* obese mice, Li et al. developed a mouse model in which ALCAT1 was ubiquitously knocked out [97]. Interestingly, ALCAT1-KO mice were protected against HFD-induced obesity and insulin resistance and displayed an increase of (18:2)4 cardiolipin in the heart. They further demonstrated the preservation of mitochondrial function in hepatocytes from ALCAT1 knockout mice [149]. The proposed mechanism involves an activation of ALCAT1 by dyslipidemia, which induces a remodeling of acyl chain of cardiolipin during HFD, leading to mitochondrial dysfunction and increased production of ROS, which contribute to the development of metabolic syndrome. In this context, loss of function of ALCAT1 preserves mitochondrial function and increases energy expenditure, which protects against HFD-induced obesity and its deleterious consequences.

Knockdown of *Taz* in mice also resulted in an increase of energy expenditure and protection against HFD-induced obesity [150]. As expected, decreased expression of *Taz* induces changes in cardiolipin remodeling and in particular a decrease of (18:2)4 cardiolipin, leading to an impairment of respiratory complexes assembly and function in the heart and skeletal muscle [150,151]. When using fatty acids as substrates, coupled and uncoupled mitochondrial respirations were increased in these mice, thereby contributing to the elevated energy expenditure. In particular, targeting the liver or pancreas may be an interesting option as the knockdown of *Taz* in β-cells and the liver protects against HFD-induced obesity without visible severe dysfunction in these organs [152].

These two studies suggest that targeting cardiolipin remodeling could be an option to increase energy expenditure and treat obesity. Counterintuitively, our recent work suggests that targeting cardiolipin content in skeletal muscle could also be an option to achieve this aim.

Indeed, we recently identified a new pathway involved in the regulation of cardiolipin content that could be of interest for the management of obesity and metabolic syndrome [46]. We identified that *Hacd1*, a gene involved in the synthesis of very-long chain fatty acids (VLCFA), i.e., fatty acids ≥ 18 carbons, is required for proper mitochondrial phospholipid content and mitochondrial efficiency specifically in skeletal muscle. More precisely, HACD1 loss of function leads to a two-fold reduction in mitochondrial phospholipid content, where cardiolipin content is the most reduced one, and without changes in cardiolipin remodeling or peroxidation. Alteration of mitochondrial membranes composition is associated with a reduction of mitochondrial coupling, which drives an increase of energy expenditure that protects mice against HFD-induced obesity and its associated deleterious consequences. Importantly, enrichment of mitochondria isolated from *Hacd1*-KO mice with cardiolipin rescued mitochondrial coupling, in vitro demonstrating a key role of cardiolipin in mitochondrial coupling efficiency in muscle [46,153]. Targeting of other genes involved in VLCFA synthesis confirmed the interest of its modulation in metabolic syndrome, although mitochondrial efficiency and cardiolipin content were not investigated in these studies [154,155,156,157,158]. Altogether, these data suggest that modulating the VLCFA elongation pathway in muscle may modulate mitochondrial efficiency and increase energy expenditure in a safe way to treat obesity.

## 5. Conclusions

In conclusion, characterization of cardiolipin content and composition is an emerging field of investigation that has started to characterize the involvement of obesity-driven cardiolipin alterations in mitochondrial dysfunction and progression of metabolic syndrome. Further exploration of this field will provide important clues regarding the exact pathophysiological mechanisms of this vicious circle. In parallel, strategies aiming at modulating cardiolipin have emerged as a promising opportunity in the management of obesity. In a logical way, several studies suggest that promoting cardiolipin content, in link with or independently of mitochondrial biogenesis, or restoring cardiolipin acyl chain remodeling or integrity towards oxidative stress could help counteract the deleterious effects of obesity. Although counterintuitive at first glance, modulation of cardiolipin content or remodeling could also be used to reduce mitochondrial efficiency and drive an increase in energy expenditure. Precise control and targeting of this modulation will be a challenge for the future to treat obesity and its associated comorbidities in a safe way. Given that cardiolipin acyl chain composition has been shown to be sensitive to lipid environment [78], an adapted diet may also drive a specific remodeling of cardiolipin to modulate mitochondrial coupling efficiency and energy expenditure. Investigation of cardiolipin alteration or modulation in obesity will be a fascinating field of research for the next years and it will be served by the growing capacities of lipidomic approaches to analyze its precise content and composition. Hopefully, cardiolipin modulation will in turn be one of the available tools for the necessary fight against obesity.

## Figures and Tables

**Figure 1 biology-11-01638-f001:**
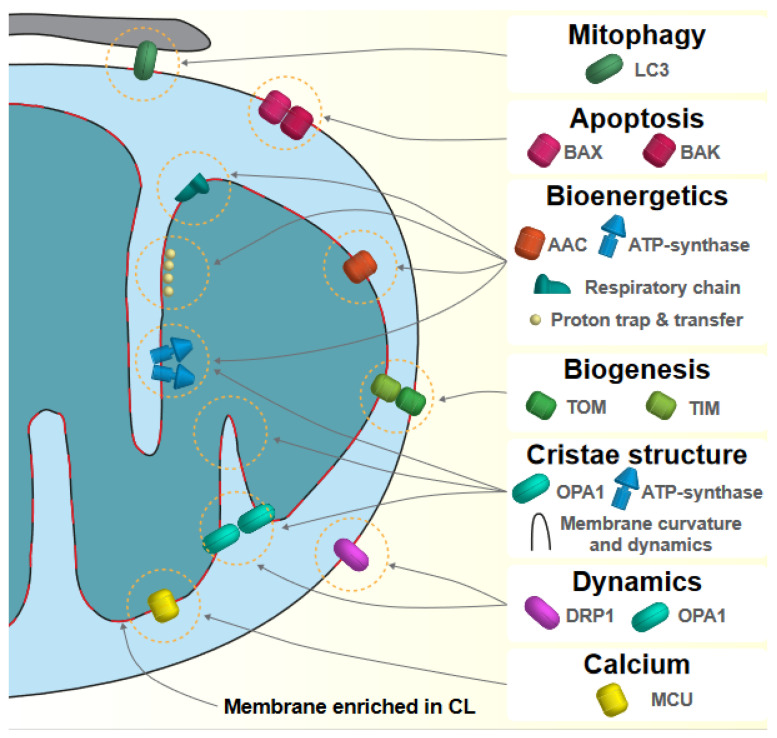
Main processes and proteins regulated by cardiolipin. *Mitophagy and apoptosis*—Translocation of cardiolipin (CL) to the mitochondrial outer membrane is involved in signaling the execution of mitophagy (in interaction with LC3) and apoptotic cell death (in interaction with BAX and BAK). **Bioenergetics**—CL is essential for mitochondrial ATP production through the regulation of ADP/ATP carrier (AAC) activity, the assembly of ATP-synthase and respiratory complexes and super complexes. In addition, CL has been shown to interact with protons and to facilitate their transfer from the respiratory chain to the ATP-synthase for optimal coupling. **Biogenesis**—Mitochondrial biogenesis relies on the incorporation of mitochondrial proteins through TOM and TIM translocases, whose proper function necessitates CL. **Cristae structure**—CL enrichment in cristae and its interaction with proteins involved in cristae structure such as OPA1 and the ATP-synthase confer specific physical properties to the inner membrane and favor its strong curvature, which optimize mitochondrial ATP production. **Dynamics**—CL influences mitochondrial dynamics through its interaction with GTPases involved in mitochondrial fusion or fission, such as OPA1 or DRP1. **Calcium**—Mitochondrial processes necessitate a tight control of calcium concentration, notably by the mitochondrial calcium uniport that interacts with CL. Parts of the membranes that are enriched in CL are represented in red.

**Figure 2 biology-11-01638-f002:**
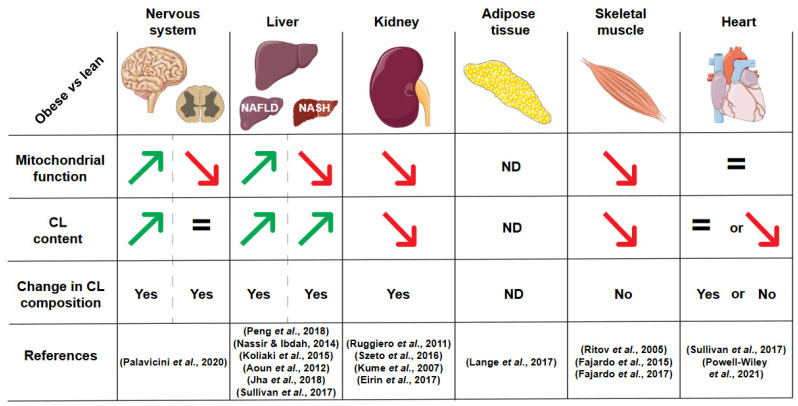
Summary of published alterations of cardiolipin in obese patients or rodents. CL: cardiolipin, NAFLD: non-alcoholic fatty liver disease, NASH: non-alcoholic steatohepatitis, ND: not determined. References, in order of citation: (Palavicini et al., 2020) [5], (Peng et al., 2018) [6], (Nassir & Ibdah, 2014) [91], (Koliaki et al., 2015) [92], (Aoun et al., 2012) [93], (Jha et al., 2018) [95], (Sullivan et al., 2017) [96], (Ruggiero et al., 2011) [9], (Szeto et al., 2016) [10], (Kume et al., 2007) [100], (Eirin et al., 2017) [101], (Lange et al., 2017) [112], (Ritov et al., 2005) [8], (Fajardo et al., 2015) [121], (Fajardo et al., 2017) [122], (Powell-Wiley et al., 2021) [103].

## Data Availability

Not applicable.

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
