# Peer review of "Cardiolipin Alterations during Obesity: Exploring Therapeutic Opportunities"

_biology, 2022, doi:10.3390/biology11111638_

Round 1

Reviewer 1 Report

The review describes a possible association of cardiolipin metabolism with obesity.

However, I must say that the authors' arguments that cardiolipins play an important role in the manifestation of obesity look rather weak, and the review needs a higher level of argumentation. Indeed, with obesity, the entire lipid metabolism changes, including the metabolism of cardiolipins, but the axe: the causal relationship (chicken-egg) should be expressed in more obvious forms than the description of what happened in models of obesity, for example, such as feeding with a high-lipid diet. I think that the etiology of obesity (in a real situation, not in models) and related phenomena, such as various kinds of stress, should be put in the first place, where cardiolipins should be proven as mandatory elements.

I am surprised that the authors in their review did not find a room to describe the works from the laboratory of V. Kagan from U. Pittsburgh, VA, who, in my opinion, are among the oldest and most influential researchers of cardiolipins. As one might note, given that there are a huge number of cardiolipins molecules (hundreds), I would prefer to use the plural to refer to cardiolipins.

It seems to me that one very short phrase about the protection of cardiolipins from oxidative stress with reference [88] is not enough and a sufficiently broad description of this topic is required, indicating the possible oxidation of double bonds of cardiolipins associated with changes in the ion-conducting properties of the inner mitochondrial membrane, leading to pathologies, including aging, which can be mitigated by antioxidants in addition to citation of relatively recent publication [10] (for example, see for review doi:10.1016/j.bbabio.2010.03.015 based on earlier publications by Antonenko et al., Biochemistry (Moscow) 73 (2008) 1273-1287). This is on the one hand. And on the other hand, another antioxidant effect of cardiolipins should be mentioned, for example, consisting in providing cytochrome c with peroxidase activity, which is effectively surrounded by cardiolipins molecules (again, see V. Kagan's work). In this aspect, it is important to consider the work cited by the authors [121], which demonstrates a decrease in cardiolipins in obese patients, but again, given the huge variety of CL, I prefer better to see which cardiolipins decrease, whether their double bonds are oxidized, whether lipid peroxides of CL are formed, etc. The authors should show their strong attitude to the relations of CL to obesity rather than describe the phenomenology of associated events.

Minor comments

More recently, another perspective was considered, which places mitochondria are (???) the heart of energy balance a…

Mitochondria are major actors in energy production, ion homeostasis, free radicals’ production, and apoptosis, working as a signaling platform…. (literally speaking, energy production and ion homeostasis do not belong to the signaling systems).

Reviewer 2 Report

The manuscript deserves to be accepted after minor revision, provided that this conclusion is shared by the editorial committee. The careful reader of this article might appreciate finding in the final version of the article an answer to the  following questions:

Can the problem associated with cardiolipin remodeling be resolved through the transgenic expression of an appropriate enzyme? In the agouti model of type 2 diabetes, over-expression of catalase protects cardiomyocites contractility (Diabetes 53; 1336).

Phospholipids generally are dimeric. Can the authors provide a plausible explanation for why cardiolipin consists of four fatty acyl acids?

According to the authors, cardiolipin remodeling is a failure of evolution or an adaptation?

Round 2

Reviewer 1 Report

The authors have responded to most of my comments. Only one remark was ignored about which of the many forms of cardiolipins respond to external and internal challenges, including obesity, so I would recommend that the authors still work to include such a discussion in the manuscript, even if such data are not presented in the world literature. It is necessary to emphasize that there are many forms of this lipid and try to discuss why this diversity takes place and find possible relevance to obesity. 

Round 3

Reviewer 1 Report

I think that after the second interaction, the manuscript sounds better and I am satisfied with the authors' reply.